# When the Normative Is Formative: Parents’ Perceptions of the Impacts of Inclusive Sports Programs

**DOI:** 10.3390/ijerph191710889

**Published:** 2022-09-01

**Authors:** Jason Rodriquez, Anika Lanser, Holly E. Jacobs, Ashlyn Smith, Sharbari Ganguly

**Affiliations:** 1Department of Sociology, University of Massachusetts Boston, Boston, MA 02125, USA; 2Center for Social Development and Education, University of Massachusetts Boston, Boston, MA 02125, USA; 3Special Olympics, Inc., Washington, DC 20036, USA

**Keywords:** developmental disability, high school, inclusive sports, intellectual disability, parents, Special Olympics, Unified Champion Schools, Unified Sports, qualitative

## Abstract

This qualitative study examines the perspectives of parents of youth with intellectual and developmental disability (IDD) who participated in Special Olympics Unified Sports^®^, a program in which high school students with and without IDD compete on the same team. Based on semi-structured interviews with parents (*n* = 23) as part of a larger evaluation of Special Olympics Unified Champion Schools in three states in the United States, thematic analysis shows that parents perceived improvements in their child’s social and emotional skills, including demeanor and attitude, an enhanced sense of belonging at school, the emergence of new friendships and social opportunities, and rewarding interactions that flowed from the opportunity to engage in normative activities. Implications for schools and families are discussed in terms of the importance of providing and facilitating meaningful opportunities for inclusive extracurricular activities such as sports for youth with IDD.

## 1. Introduction

Inclusive education refers to a philosophy that informs school-based practices intended to promote equity, acceptance, and the feeling of belonging among youth with disabilities [1]. The implementation of inclusive sports programs is a key component of inclusive education more generally, especially since participation in such extracurricular activities has a wide range of positive impacts on children, including improved health, overall development, and well-being [2,3,4]. For those with intellectual and developmental disability (IDD), research has shown that participation in sports provides benefits in the areas of self-esteem and self-confidence [5,6,7,8]. Furthermore, sports participation among youth with IDD also has a positive impact on friendships, social competence, and enhanced opportunities to participate in the same activities as their peers without disabilities [9,10,11]. Clearly, sports have the power to enhance lives, especially among adolescents with IDD. However, they often lack the same opportunities to play sports and participate in other social-recreational community activities compared to children without IDD [11,12,13,14]. Although typical barriers to sports participation, such as time, cost, and transportation, exist for youth with and without IDD, those with IDD face additional barriers that are largely the result of misperceptions about the accessibility and safety of sports for individuals with IDD and attitudinal barriers surrounding the ability of those with IDD to effectively participate and compete [12,15,16,17].

The fewer opportunities and heightened barriers that youth with IDD face reflect and reinforce the persistent marginalization that limits their chances for social and emotional development that is generally available to their peers without IDD. This is particularly important regarding school-based sports programs, which provide structured opportunities to develop friendships with peers and positive relationships with adults, such as coaches and teachers. Children with IDD are often excluded from these opportunities, limiting their chances to socialize and participate in meaningful activities that are readily available to their peers without IDD, such as the feeling of belonging on a team that shares and works towards a common goal [18].

For youth with IDD to realize the full benefits of sports, opportunities to participate, and participate meaningfully, must exist. Research suggests that these opportunities, when available, are often facilitated through organized activities that explicitly target youth with IDD [19,20], such as through Special Olympics. There is overwhelming evidence of the opportunities and benefits that Special Olympics provides for individuals with IDD on perceptions of competence, self-esteem, quality of life, psychological well-being, and social networks [5,6,9,21,22].

Twenty-five years ago, Special Olympics expanded its traditional sports program to include Unified Sports^®^, a program where people with and without IDD compete and train together on the same team. Unified Sports is implemented in communities and schools nationwide as one component of Special Olympics Unified Champion Schools (UCS)^®^, a school social inclusion program aimed at creating more inclusive school climates by bringing students with and without IDD together through a variety of in-school and extracurricular activities [23,24,25]. Unified Sports, when implemented as part of UCS, continues a decades-old movement to promote inclusive cultures in schools and communities.

Promising evidence suggests positive impacts of participation in Unified Sports for youth in the areas of personal growth, cross-disability friendships, belonging, and increased confidence [26,27,28]. Unified Sports athletes with IDD also associate social inclusion with the friendships formed as part of the team experience. This reinforces the idea that meaningful connections between those with and without IDD are the key to moving beyond just participation and towards the creation of deeper connections between participants [28,29,30].

While examining the benefits of participation in sports for those with IDD themselves is certainly important, it is also important to understand the perspectives of other stakeholders to provide a more holistic understanding of the overall impact. From a socioecological perspective, family support is a primary influence on the positive sports experience for youth [31]. For Unified Sports, the parents of student participants are ideally situated to provide insights into the benefits of the program, as they can make holistic assessments based on their intimate knowledge of their own child. Among children with IDD, families play a key role in facilitating, or inhibiting, their participation in sports programs [13,32,33,34].

Parents of youth with IDD who do facilitate their child’s participation in sports often see it as having a positive impact on their child’s social and emotional development. Research has shown that these positive impacts are driven by the opportunity for their children to participate in more normative sports opportunities that are offered to their peers without IDD [9,35,36]. Parents of children who participate in Special Olympics, for example, report that when their children participate in sports, they see improvements in their children’s independence and autonomy [37]. Furthermore, opportunities to spend time away from family and attend competitions provide opportunities for greater independence and autonomy to emerge. These parents report improvements in their child’s self-concept through opportunities to set and accomplish goals, demonstrate their abilities, and receive recognition for success [37]. In addition to increased autonomy and independence, research has shown that parents of children who are in Special Olympics believe that participation in sports provides improvements in self-esteem and confidence [38,39], as well as emerging friendships and the opportunity to form meaningful peer interactions [9,27,40]. Given the value of parents’ perspectives for learning about their child’s participation in Special Olympics more generally, what might we learn from parents about the benefits and value of Unified Sports?

This study takes the existing scholarship one step further by examining parents’ perspectives of Unified Sports participation for youth with IDD who participated in Special Olympics Unified Champion Schools (UCS). Based on 23 interviews, this study sought to understand how parents perceived their child’s participation in, and the impacts of, inclusive sports for their children. Parents reported positive changes in their child’s demeanor and attitude, an enhanced sense of belonging within the school, new friendships and opportuniti10es, and the value of opportunities for their child’s participation in normative activities often unavailable to children with IDD. In the discussion and conclusion, we explore the implications of these findings for families and schools’ implementation of inclusive sports.

## 2. Materials and Methods

### 2.1. Participants

Participants in this study were family members (n = 23) of high school students with IDD participating in Special Olympics Unified Sports through their school’s UCS program. The family members were recruited as part of a larger national evaluation of UCS’s implementation and impact at the high school level. The Unified Sports that students played as part of UCS included flag football, basketball, soccer, bowling, and bocce. More detailed descriptions about UCS and Unified Sports in high schools can be found in Citations [38,41].

Table 1 presents the relationship of the family members to the children as well as information they provided about their child’s primary disability diagnosis. These family members were parents of children from six high schools in three states in the east, Midwest, and mountain regions of the United States (see Citation [38] for additional details about the schools and their UCS programs). Nineteen out of twenty-three of the family members interviewed were the child’s mother, three were the child’s father, and one was a grandparent who was the child’s legal guardian. Given this sample, for the sake of simplicity, we typically refer to the whole sample as “parents” or “family members”, and we note the grandparent specifically in the results when we quote that individual. Most of the students were male, and the average age was 17. The students were evenly split between underclassmen and upperclassmen and evenly split in terms of the severity of disability as reported by the family member interviewed. For all students, this was the first year their high school had implemented Unified Sports and UCS and was thus their first opportunity to play inclusive sports in their high school. However, about half the students (*n* = 12) were involved in Special Olympics sports prior to joining the new Unified Sports team at school.

### 2.2. Procedure

This study utilized a qualitative approach, specifically interviews of family members of children with IDD. Interviews were chosen because they placed the family members’ voices at the center of the narrative, allowed them to share stories about their child in their own words, and provided rich information about each child and family’s unique experience. Interviews were conducted during end-of-school-year visits to the high school as part of an evaluation to assess the impact of UCS implementation. Approximately one week after the school visits, family members of students on the schools’ Unified Sports teams were contacted over the phone by a member of the research team, using the phone number provided on the study consent form that family members signed at the beginning of the school year. Voicemails were left for family members who did not answer the initial call to conduct an interview; on average, family members were contacted three times before an interview was scheduled and completed. Conducting the interviews with family members spanned an approximately six-week period.

The interview protocol was semi-structured and designed to elicit perceptions of their child’s social experiences at school in relation to its UCS program, including Unified Sports. The protocol was developed by the research team for this specific purpose, and the questions were based on similar interviews with other parents who had children participating in UCS prior to this study. At the time these interviews were conducted, UCS was called Project UNIFY, and when participants referred to Project UNIFY, we stayed true to the quotes and did not remove or change them, although they refer to the same program now known as UCS. The protocol included questions related to inclusion and belonging in the school community, social interactions with peers and adults in and out of school, participation on sports teams in and out of school, and the impact of participation in sports on the child’s school experience and personal growth. See Table 2 for the interview protocol questions most relevant to this study. Interviews lasted an average of 29 min, totaling 11 h and 4 min. All interviews were audio-recorded and professionally transcribed. All procedures and data collection instruments were reviewed and approved by the University of Massachusetts Boston IRB.

### 2.3. Data Analysis

Transcripts were analyzed following the principles of thematic analysis [42]. Thematic analysis was chosen because it is a flexible strategy that allows for an inductive, data-driven approach to interpreting qualitative data [43]. After an initial review of all transcripts, two members of the research team developed an initial codebook. Three researchers, including the two who developed the codebook, then used this codebook to code a randomly selected subset of transcripts. Then, the researchers reviewed each other’s coded data for points of convergence and divergence, providing an opportunity to iteratively revise the codebook through group discussion and the rereading of the transcripts until a consensus was reached. As part of this process, we added codes, combined codes, removed codes, modified codes, and provided labels and examples for codes to exemplify their conceptual meaning. Once the adaptations were complete, two members of the research team recoded the initial selection of transcripts with the adapted codebook. At that point, there was a high degree of consistency between the two coders, and they agreed that data saturation had been reached, meaning no new codes were necessary and no new themes emerged [44]. The first draft of the codebook had eleven thematic codes, which became seven thematic codes in the final draft. Over the course of this revision, five existing codes were combined into two new codes, three codes were removed, one code was added, and three codes were modified. For example, Parent awareness of the student’s academic program and Parent involvement in the student’s academic programming became Parent perception of schools with subcodes that addressed advocacy and support. See Table 3 for the frequencies of thematic codes that centered the analysis. The table reflects how the data were coded during the analysis that led to this article. The codebook was considered final, and the remaining transcripts were coded using this final version of the codebook. This was followed by review and discussion of coded data within each code to confirm consensus with the larger research team and assess theme representation [45]. QSR International’s NVivo software was used for coding and analysis, which included querying transcripts, viewing total occurrences of each code, and viewing coding matrices.

## 3. Results

### 3.1. Changes in Students’ Demeanors and Attitudes

Overall, parents indicated that their child enjoyed Unified Sports and that it supported their child’s development. For example, one parent said, “UNIFY has definitely been a very wonderful experience for us”, while another similarly noted, “I think it’s a great program and he really enjoyed it”. Parents positively assessed the benefits of the program, with one elaborating that “I just like it because I think this is the best thing that’s ever happened to him…this Unified Sports is just fantastic. It feels real, it doesn’t feel contrived”.

Parents reflected on the ways that their children’s attitudes towards themselves, sports, and school improved because of their participation in UCS or in other inclusive sports programs. They described an increase in their child’s confidence, feelings of excitement, and motivation to attend school. For some students, parents said higher self-confidence came from the emotional investment in being part of a team. For example, one parent reflected, “It gave him a little bit more confidence…It was a really big deal to him and a happy memory to get to play on the team”. Another parent spoke about the positive impact of having the girls’ basketball team that her daughter managed show support for her daughter at Unified basketball games. “Her confidence has gone leaps and bounds”, the parent explained, “I feel like the UNIFY project has been wonderful for [my daughter]. It’s given her a lot of self-confidence”. Another parent compared her daughter’s previous negative social experiences with her perspective after joining the team, “Now that we’re up here in [High School], her attitude has changed. She loves going to the school and her self-esteem has improved a lot”. The team provided a space for students to expand skills that made them feel more confident and build a sense of belonging to the team and the larger school community.

In addition to confidence, parents noted additional benefits of UCS for their child’s development. One explained that it is “not just the sport, but the character-building, and the leadership, and other things”. Along these lines, another parent reflected on how the program impacted their son’s consideration for others: “He’s always been a compassionate little guy, but he seems to have even more compassion for kids with actually more needs than him…I think it’s had a very positive impact on him”. Sports became a significant part of students’ experiences at school, and with participation came the opportunity for new social experiences.

Parents also perceived their children to be increasingly excited to participate in sports programming and happier because of it. Parents saw this excitement emerge when children approvingly spoke about their Unified Sports experiences. One parent shared, “He’ll try to engage us in conversation. He won’t really talk about things that he does outside of what we know, unless it’s something that’s really significant to him. He talks about it a lot, and now he’s all excited about soccer”. Another described their daughter’s attitude towards the team as “Wildly excited to go. Wildly excited to do the sporting events. For the first time in her life, she is like hanging that t-shirt up. Those things have become so important to her”. One parent put it simply, “She’s happier when she’s playing”. Parents described their children as excited to participate and independently invested in being on the team. In some cases, the increased excitement students had regarding their participation in sports also reflected an increased sense of belonging in school, a theme explored in greater detail below.

### 3.2. Enhanced Sense of Belonging

In addition to the positive changes in demeanor and attitude observed, parents also explained that UCS created a more positive school environment for their children that generated a greater sense of belonging to the school community. For example, one parent said that because of the inclusive sports program, their son “has something to look forward to. He’s very excited about something he can participate in. It kind of makes him feel like he belongs, like part of the school outside of just going to school” and added that “it’s made [school] a lot more positive for him, and it’s been a very good experience for him”. Another parent said that at their son’s school, “there’s not a lot of activities for kids with autism” and that Unified Sports “has been more of a positive” and that other students are now “interested in saying, ‘Hi’ and being welcoming to him…it was nice to have that”.

Parents emphasized that UCS enhanced their child’s sense of integration at school. For example, one parent explained that the program made her daughter “feel a more concrete part of the school” and that “It has made her feel like that’s her *school*, not that’s my *classroom* (emphasis added)”. This quote suggests that because of UCS, the student felt a deeper sense of belonging and spirit towards the school rather than simply having a classroom that they go to for instruction. Even though their school had “made huge efforts with the peer mentor program”, there was something unique about participation in sports because “they feel like I’m not just in a program housed at a high school, I’m actually an important part of that school”. Referring to the school mascot, she said her daughter “definitely feels like a [High School] Eagle” and “they want to come cheer their school on, which I love”. Another parent said that the program made their son “feel that he was a part of something” by having to go to practices and take team buses. The parent explained, “He used to kind of want to miss school a lot, and this year he hasn’t wanted to miss a day…it definitely made his school experience a lot more enjoyable”, adding later for emphasis, “it made him want to go to school every day”. Another parent noted that their daughter’s school’s inclusive athletic program and “more after-school type activities has helped her feel more included, and motivated too”.

Some parents also noted that the program positively impacted students without disabilities who participated, further enhancing a sense of belonging at school. “I think it’s probably just as good for the typical kids as it is for the kids in special ed”, reflected one parent, “because they’re going to see that and understand really what life’s about”. Another parent suggested, “It’s been good for both sides”. Along with the benefits for students’ sense of self and belonging, playing on Unified Sports teams improved teamwork and leadership skills while also creating new social opportunities for students to make friends of all abilities.

### 3.3. New Friendships and Social Opportunities

Parents spoke about Unified Sports as one of few opportunities for their child to socialize with peers in ways they otherwise would not have been able to if not for the inclusive team context. A parent reflected, for example, on how sports shaped his son’s social experiences: “He did connect with some kids he might not have connected with had he not played”. Other parents saw similar benefits to playing on the team. “He made some good friends. He really enjoyed playing”, said one parent. Another said of Unified Sports, “he had a good time and made some friends”. Parents saw these emerging friendships as a beneficial aspect of Unified Sports.

Parents also saw the benefit of the unique social setting of sports teams and noted impacts on their child. “I think she feels that camaraderie. I think she really likes it. I mean she’ll talk about it a little bit, ‘My friends play basketball.’ She’ll try to engage us in conversation”, one parent reflected. Another parent spoke about their daughter learning to work strategically with teammates. They explained, “Anytime [my daughter] would be dribbling the ball and coming down, [her teammate] would park right underneath the basket a little bit short because she knew [my daughter] would [make] the perfect pass to the other girl and then she would just turn and put it up”. A third parent noted the increased positive attention their child and others received when participating in Unified Sports, reflecting that “Some of the kids out there, that was their first time in the spotlight…to see them light up was just absolutely amazing”. According to parents, their children were not just excited about sports; they found value in the friendships they made with teammates. For example, a parent reflected, “I just like the way he looks when he’s playing with the guys, he thinks that he’s part of the team. I don’t know how to explain it, but it’s one of the best experiences he’s had”. The same parent noted that “It was the first for him to go and be without us watching him, or without somebody keeping an eye on him”. Playing on Unified Sports created new social opportunities for students to make friends of all abilities and explore friendships on their own terms.

Extracurriculars in general often serve as important spaces for socialization with peer groups. Without inclusive extracurricular activities available, some parents struggled to help their children socialize. One parent, for example, spoke about how the lack of available inclusive activities drives them to seek out any and all opportunities, saying, “There’s almost no extracurricular things for [my child] to do, so anytime there is something [such as Unified Sports], we take advantage of it”. Unified Sports helped ease the burden on parents to create these social opportunities. The same parent explained, “I think it has helped because it gets him out of the house. It gets him being social”. Another parent saw the social role of Unified Sports similarly: “It’s something that he enjoyed doing a couple times a week and seeing his friends. It’s a good social activity. It’s a good way to build relationships”. Reflecting on when their child spends time with friends, one parent responded, “Mainly through Special Olympics sports. They’ll have practices and meets for track, and then she’s in the bowling league so she is bowling every Sunday with them”. Not only do school-based sports serve as social time for students, but they also represent one of the only times students see friends outside of school. Sports activities and teams help students to build new social connections, but the schedule has its own benefits. The regularity of practices and games supports students in maintaining and building upon those new friendships.

For students with many existing social connections, often built through participation in intramural sports, Unified Sports expanded social networks and provided opportunities for students without disabilities to reciprocate their friend’s support. One grandparent shared their grandson’s experience managing Varsity teams for the school, explaining, “From the first day there, he has just been scooped up and is one of the guys. He was the football manager and basketball manager both years, and then this year they created a position for him for baseball”. This led the student to develop a robust social network: “He actually has more friends in the general student setting than in his special ed class”. Through serving as a manager for Varsity teams, the student built a robust social network across the school. Still, participating in UCS allowed a new aspect of these friendships to develop. The grandparent remembered,

“When Unified Sports started, he basically ordered all of them to come to his games. He said, ‘I supported you guys, and I expect you at mine,’ and it worked. The football team, basketball team, even wrestling, who he’s not even part of the wrestling team, they have shown up at [the Unified] games to cheer him on”.

The parent of another student who served as the manager for her school’s basketball and soccer teams recalled her daughter’s experience of friendships built through sports participation and affirmed the reciprocal dimension that Unified Sports provided. She reflected,

“A couple of girls from the basketball team attended…It’s been nice that they’ve, the girls from the basketball team…She has done it for four years, so she’s more a part of the team-really supported her too. So, I think that made her feel good”.

Even for students who were already deeply socially connected through existing sports participation, being a player on the Unified Sports team added another dimension to those friendships. Students finally received reciprocal support from their intramural athlete friends and had space to establish themselves as athletes too.

Parents noted approvingly that their child benefited from different groups of people working together as part of UCS. Lauding the high engagement of teachers and staff who help implement UCS activities, a parent explained,

“I’m just really proud of the experience that he’s had, the faculty and staff are really good at what they do and they have a passion for those kids, and I’m glad that [my child] has been a part of it”.

Another noted that the program became a significant part of the school more generally, explaining, “It’s become very important to that school. Teachers, everybody’s invested in it”. Another noted approvingly that Unified Sports brought people together: “There’s the parents and the students and a group of athletes that work with it, and then some of the administration have all worked together on this whole thing”. A few parents mentioned that UCS affected them personally. One parent noted, “it was a very good experience for both him and me”. Another parent said they were inspired to participate: “Watching the kids compete, it wasn’t even [my child] that made me bowl. It was the other kids…And just all the volunteers there and—so just an awesome experience”. When parents, teachers, and administrators work together to support opportunities for inclusive activities for students with IDD, their engagement pays dividends to all involved and the entire school community.

### 3.4. Participation in Normative Activities

Though parents often named specific impacts they attributed to their child’s participation in sports, such as increased self-confidence or newfound social connections, many also described the kinds of experiences their child had as part of the team, such as on the way to the game or at practice, as impactful. Beyond the direct benefits they saw for their children, parents were pleased about how participation in Unified Sports provided the opportunity for their child to have the same normative experiences as their peers without IDD. Parents’ reflections emphasized how inclusive extracurricular opportunities could mean that taking the bus with teammates to an away game, and all of the informal socialization and social-emotional learning that comes with it, became accessible for the first time. One parent shared, “This is the first time he’s ever been on a basketball team, and that’s probably one of his, besides football, it’s one of his favorite sports”. A different parent explained, “She had never bowled before. So it had made her be able to bowl for the first time”. Another parent reflected,

“So every Saturday morning she would get up, get her t-shirt and be ready to play basketball. And it was so cool…the kids were so excited, because normally when they do things, the whole school’s not there; everybody’s not cheering; it’s not like a real sporting event, and this is like they were participating in a real sporting event”. This parent added that they liked that their child’s team was “going to have a practice during the week, which makes them like all the other athletic organizations”. Parents saw participation in the program itself as a benefit to their children because it allowed them to participate in activities that are typical for high school students.

Parents spoke positively about the competition and “realness” of Unified Sports, like intramural or interscholastic sports. One parent remarked, “I think that while he really enjoys being a student manager for the Varsity sports, I think it leaves a big hole in his desire to still compete”. Another emphasized, “It makes the high school experience so different and so much more full and so much more like a real high school experience”. For students who love sports and who previously lacked opportunities to compete, these aspects represented more salient parts of their experience because they mirrored the sports experiences of their peers without IDD. For some students, participating changed the way they conceptualized their identity. One parent noted, “She sees herself now more as an athlete, where before she didn’t”. For others, it developed an interest in playing a sport at the intramural or interscholastic level. “I think next year she’s going to try to go out for the regular basketball team and stuff”, one parent said.

Parents also saw an impact in moments that were emotionally or socially challenging for their child. Quintessential to playing sports is responding to circumstances beyond one’s control, such as an unfair call or a lost game. In sports and in life, individuals engage with others whose goals and actions can be at odds with their own. Parents reflected on times when their child navigated challenging situations in sports as beneficial for their personal growth and something they appreciated about the context of Unified Sports. The competitiveness of the program presented unique opportunities for growth that their children did not often have despite their relative prominence in the lives of children without disabilities. One parent described Unified Sports as giving kids with IDD “a little bit of ownership, so I just really like it. I like the fact that sometimes the teams lose, sometimes people lose”. This gives students an authentic experience where “they deal with it just like how you’re supposed to”.

Another parent recalled the time their student was reprimanded by a coach for being too aggressive. “He had tried to go and calm himself so he went at the end of the bleachers and was trying to breathe through it, but he just was too mad, he couldn’t come back from that rage kind of setting. And so, we let him try and process that for a couple of minutes”. It was an opportunity for the parent and the child to work together to learn “what was going on from the other perspective” and to parse the difference between “what [coach] probably meant when they said aggressive, versus what [my child] interpreted that to be”. This allowed the student to practice self-regulation in a new context and receive support to resume participation. Another parent told a story of having to check her own “Mama bear” reaction when her son told her that his coach “threatened to kick him off the team” and that “if he didn’t straighten up his act, he wasn’t going to the [professional baseball] game”. Despite the parent’s initial inclination to call the coach, her son responded, “No mom, I took care of it. I was out of line. I did snap at [coach] and I apologized to him. And it’s okay now”. When faced with a new perspective on their actions, the student was able to address the conflict, come to an understanding about how their behavior impacts the team, and resolve their relationship with the coach on their own. Through sports, students with disabilities gained access to environments where these social and emotional skills are learned. Inclusive sports programming can broaden the contexts where students with IDD have access to these normative developmental experiences.

Parents expressed gratitude for these aspects of the program and the impacts of sports participation. They often reflected on ways they would like to see Unified Sports expand. Some parents wanted more opportunities to engage with sports in ways that more closely mirrored the structure of intramural or interscholastic sports. One parent reflected along these lines: “[To] have one school’s team against another school’s team would be kind of neat”. Parents also wanted to see a broader range of inclusive opportunities available, both within sports and beyond. One parent noted, “It’d be nice to have more options for [Student]. Like basketball, if he wanted to play basketball he could”. Another advocated, “I’d kind of like to see them use the UNIFY model and pull it out a little bit as far as maybe the drama program and the arts”. Parents were therefore not just invested in their child’s opportunity to compete on a sports team and the accompanying developmental impacts but also advocated for greater diversity of extracurricular activities, affirming the importance of inclusive activities at school as crucial normative experiences for children with IDD.

## 4. Discussion

The purpose of this study was to center the parent perspective to advance our understanding of the social and developmental impacts of inclusive sports programs on youth with IDD. While parents noted specific positives for their children, they also found the existence of the program itself and the normative experiences provided through Unified Sports to be distinctly beneficial. Parents reported that UCS provided children with a more robust school experience, created social connections, and provided the same social and emotional learning opportunities as their peers without IDD. The parent’s perspective provides a unique position from which to further understand the ways that sports programs benefit the children who participate. These findings have implications for the social and developmental improvements that inclusive sports programming can provide for children with IDD and their families.

First, these findings have implications for how researchers and educators consider the social significance of inclusive sports, which can inform program development and implementation. Consistent with previous research, parents mentioned new friendships and meaningful social connections that occurred through their child’s experience in Unified Sports [16,18,21,35]. Our findings take this one step further by hearing from parents directly about the role of school-based inclusive programs in mitigating traditional barriers that prevent youth with IDD from participating in sports alongside their peers. This approach reveals the benefits to students from a new perspective, illustrating how inclusive programming provides opportunities for students to have the same developmental experiences as their peers without IDD.

Our findings add additional support to the evidence base suggesting that participation in Unified Sports supports students’ self-esteem and confidence [38,39] while also positively impacting opportunities to develop teamwork and leadership skills, all of which contribute to enhanced social-emotional skills. For example, parents’ perspectives on the most salient social and emotional benefits of sports participation show their child to be practicing increased personal responsibility, self-management, self-awareness, and investment in being part of a team.

Furthermore, students’ increased sense of belonging to both the team and school because of Unified Sports and UCS has implications for educators’ and schools’ implementation of inclusive programs. Parents described their children as newly excited to go to school and to be part of a team and “belong to something”; students found deeper personal connections with their school, teachers, and coaches. This finding is consistent with previous studies that attribute an increased sense of belonging in school to participation in extracurriculars such as inclusive sports programs [46]. Parents credited students’ newfound motivation to go to school to their involvement on the Unified Sports team, as they were excited about being on the team and the feeling of camaraderie that came with it. When spaces exist to engage with peers through extracurricular activities outside of the classroom, students with disabilities can begin to recognize and feel that they are an important part of the school [47].

Parents also saw their child’s increased sense of belonging as a function of the program’s effects on students without disabilities. Although parents saw many impacts on their own child, they also described the changes they saw in children without IDD over the course of the season. Parents described peers without IDD as becoming more inclusive of students with IDD and developing new perspectives. Inclusive sports programs such as Unified Sports foster inclusive school cultures and lead to an increased sense of belonging among students with disabilities [27,28]. As schools use programs such as UCS to foster inclusion and belonging, educators can build on program impacts by creating more independent but structured social spaces for students to build friendships across disability. The benefits of belonging come from the experience of competing on the team but also from the moments in between where students can informally engage with each other and build meaningful connections. To further their goals of inclusion and belonging, schools must adopt a range of inclusive activities that provide more opportunities for students to extend their social worlds and practice key skills in a new context.

Finally, the opportunities for the normative experiences that inclusive programming provides have implications for how parents, schools, and communities come together to meet the needs of students with IDD. Most of what parents described as some of the most formative experiences for their child were experiences that occur often for children without IDD, such as taking the bus to an away game or hanging out with friends after practice. For some students with IDD, this was their first opportunity to compete in a sport they have loved for years. While some normative experiences were positive, students also experienced the challenges and frustration that can come with playing sports, such as losing a game or being reprimanded by their coach. Experiencing these challenges allows youth with IDD to experience personal growth and character building. The competitive nature of Unified Sports was something parents also pointed to that made their child’s experience feel more “real”. When students competed on the team, they took on challenges independently and alongside peers. Teammates and coaches relied on them. Parents and friends cheered them on. In all of these instances, students practiced important social and emotional skills in a new context, and parents were glad to see their children have experiences they viewed as authentic.

With few options for extracurricular programming available to their children with IDD, parents were especially grateful to have UCS and its activities, such as Unified Sports, available in the school. Still, parents hoped to see the program expand and suggested new sports or more alignment with interscholastic-level sports and wished the program would branch out into other extracurricular activities such as drama. Essentially, parents wanted inclusive opportunities that mirrored their child’s interests beyond sports, as are widely and readily available to their peers without IDD. The normative experiences students had because of participation in Unified Sports illustrate the importance of, and need for, widespread inclusive programming for students and families. Not only would an increased availability of school-based inclusive extracurriculars facilitate greater participation, but it would also allow all students the opportunity to pursue their passions and hobbies, finding activities that they love that will shape how they see themselves and the world. Parents’ perspectives emphasize how Unified Sports served as a source of normative experiences for their children and the importance of having access to programs that can provide them.

## 5. Conclusions

The goal of this study was to explore parents’ perspectives on the impact of Special Olympics Unified Sports participation for adolescents with IDD. Through this lens, we have extended our knowledge of the positive impact of inclusive sports participation and highlighted the unique impact of offering these opportunities within the school context. Our results underscore a growing body of work that supports sports participation for youth with IDD as having positive impacts across a range of social and developmental outcomes. When these activities are offered in schools, as the family members in our study noted so clearly, the impacts on students extend into an enhanced school experience and connection to the school. For some students, it could change their entire outlook about school and their own identities, enhancing feelings of acceptance and belonging. The goal of inclusive sports, as a component of inclusive education, is to promote equity, acceptance, and the feeling that everyone belongs, and Unified Sports clearly contributed to these goals [1].

While centering parents’ perspectives of their child’s participation in Unified Sports served to situate student experiences through a different lens, there are limitations to the current study. The small sample size of 23 family members may not be reflective of all parents of children with IDD who participate in sports, but rather reflective of the experiences of parents whose school was able to offer an inclusive sports program in the first place. Such small sample sizes have been noted in other qualitative work with families of children with IDD, and, when studies are conducted in schools, there are often small numbers of children with IDD relative to the general school population [33]. Further, demographic data were not collected for the parents, so it is not possible to draw comparisons across identity markers, making it more difficult to ascertain if the sample is broadly representative of parents of high-school-aged children with IDD. However, given the relative underrepresentation of parents’ perspectives on their child’s inclusive sports participation, this paper still provides valuable insight into the role of Unified Sports and inclusive programs such as UCS in adolescent development.

Future research should aim to support parents in facilitating normative opportunities for their children by increasing access to inclusive extracurricular programs. This could look like creating more school-based inclusive extracurricular activities or making inclusion a goal of existing school-based extracurriculars. Researchers could build on this work by triangulating the parent perspective amid student, coach, and administrator perspectives or comparing the competitiveness and impact of inclusive school-based sports programs. More generally, the power of inclusion and an increased sense of belonging fostered through inclusive sports programs such as Special Olympics Unified Sports should be taken seriously as a mechanism to improve the lives of youth with IDD. Such opportunities provide a “real” and authentic high school experience consistent with the experiences of their peers without IDD.

## Figures and Tables

**Table 1 ijerph-19-10889-t001:** Family and Student Characteristics.

Characteristic	*n* Size
Relationship to Student	
Mother	19
Father	3
Grandmother	1
Student’s gender	
Male	16
Female	7
Student Grade	
9th	4
10th	7
11th	5
12th	7
Primary Disability	
Asperger’s or autism	7
Cerebral palsy	1
ADHD	2
Down syndrome	2
Learning disability	2
Bipolar disorder	1
Developmental delay	1
Traumatic brain injury	1
Unknown/unnamed	6

**Table 2 ijerph-19-10889-t002:** Family Member Interview Questions.

Topic Area	Question
School Environment	Is [name] included in the school community?
	Do you think the school (both teachers and students) sees [name] as a contributing member of the school community? Do you think they see him/her as a valued member of the school community?
	Do you think the school can do anything more to include your child and other students with special needs in the school community?
	Do you feel that the students that attend [name]’s school are friendly to him/her? How?
	Are there ever times where you feel that other students are mean to him/her?
	Do you feel that there are adults at the school that [name] can go to when he/she needs help (in addition to help with academics)? If so, who?
Social Interactions	Does [name] have a friend or friends that go to his/her school?
	Does he/she spend time with this friend/these friends outside of school?
	What types of things does he/she do with friends from school?
	Are there things that make it difficult for [name] to be with his/her friends after school?
	Are [name]’s friends the same age and gender as him/her?
	Are they students with disabilities/in special education classes?
	Does [name] have a friend or friends that do not go to his/her school?
	How did [name] meet these friends?
	Do these friends live close by?
	What types of things does he/she do with these friends?
	Are they students with disabilities/in special education classes?
	Are there any adults that [name] feels are his/her friends? Who?
Unified Champion Schools Participation	What do you know about the Unified Champion Schools program at [name]’s school?
	Has [name] told you about his/her experiences in Unified Champion Schools? What has he/she said about Unified Champion Schools?
	Do you know which Unified Champion Schools activities has [name] been involved in?
	Do you think that participating in [insert UCS activities] at school has changed [name]’s school experience in any way? Have you seen any changes in his/her attitude? Behavior?
	Have you been involved with any of the Unified Champion Schools activities at the school? How?
	Other than Unified Champion Schools activities, did [name] belong to any other sports teams or clubs at school?
	Does he/she belong to any sports teams or clubs outside of school?
	Is he/she involved, or has he/she ever been involved, with Special Olympics (besides Unified Champion Schools)?
	Is there anything else you want me to know about [name] or the Unified Champion Schools program at [name]’s school?

**Table 3 ijerph-19-10889-t003:** Frequency of Thematic Codes.

Code	Subcode	Frequency
Impact of Child’s Participation in Sports	Changes in Demeanor or Attitude	15
	Opportunities to Engage in Normative Activities	32
	Sense of Belonging in School, Peer Group	64
	Social Connections, Relationships	32
Total		143

## Data Availability

The data presented in this study are available on request from the corresponding author.

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
