# Peer review of "When the Normative Is Formative: Parents’ Perceptions of the Impacts of Inclusive Sports Programs"

_ijerph, 2022, doi:10.3390/ijerph191710889_

Round 1

Reviewer 1 Report

Comments to the Author: Thank you for submitting your manuscript entitled “What the Normative is Formative: Parents’ Perceptions of the Impacts of Inclusive Sports Programs.” The goal of this study was to explore parent perspectives on the impact of Special Olympics Unified Sports participation for adolescents with IDD. The manuscript is well written and easy to follow. The author (s) did a good job in laying out the thermotical framework and key concepts. The major concern is cross-validation due to the nature of the qualitative study. The study only used one-time interview data, but no other source of information was used to triangulate the results of the study. Here are some specific comments.

Line 60-64, it is suggested that the author(s) add the reference/citation for both communities and schools nationwide examples.

Line 79-80. This is the justification for the necessity of the research. The author(s) should consider expanding this sentence by adding an insightful literature review regarding how families play an important role in high school students' participation in sports programs.

Line 81-97, line 157-175, since the study is lack of cross-validation, no other source of information was used to triangulate the results reported in the study, and the data analysis section is limited in detail. How many codes in the first place and how many were dropped, added, combined, or removed after the discussion, and why? It would be helpful the create a table to show the process of coding and themes from the beginning to the end. 

Author Response

Thank you for the positive comments that the manuscript is “well written and easy to follow” and that we “did a good job in laying out the theoretical framework and key concepts.”  

  1. Line 60-64, it is suggested that the author(s) add the reference/citation for both communities and schools nationwide examples. 

Authors’ Response: We added three references to evidence the point made in this sentence. Please see page 2, line 69, citations 24-26. 

  1. Line 79-80. This is the justification for the necessity of the research. The author(s) should consider expanding this sentence by adding an insightful literature review regarding how families play an important role in high school students' participation in sports programs. 

Authors’ Response: We did exactly as suggested and feel it strengthens the justification for the study. Thank you for the suggestion, see lines 79-87. 

  1. How many codes in the first place and how many were dropped, added, combined, or removed after the discussion, and why? It would be helpful the create a table to show the process of coding and themes from the beginning to the end.  

Authors’ Response: We added this additional information in section “2.3 Data Analysis” on page 5, line 184-190 and included a table of the codes we used for the analysis in this paper. We believe these additions clarify our coding and analysis. 

Reviewer 2 Report

Thank you for the opportunity to review this paper. The study involves a thematic analysis of responses gathered from semi-structured interviews conducted with 23 parents of high school students with intellectual and developmental disabilities (IDD) who participated in a Special Olympics program where those with and without IDD compete and train together. Results of the analysis showed that parents saw improvements in their child’s social and emotional skills. Overall, I believe the authors did a nice job of developing the manuscript throughout.

Did the authors develop the semi-structured interview used in this study? Is this a validated measure? Please expand to further discuss this and establish its appropriateness for this study.

How the authors arrived at their use of thematic analysis is not clear to me. Explain why this method was chosen over other available qualitative analysis approaches.

A major issue of this study the limited number of participants, which translates to limited data from which any significant conclusions can be drawn. However, as noted by the authors, given the limited number of students with IDD and underrepresentation of parents’ perspectives on their child’s inclusive sports participation, this paper does offer some contribution to the literature on this topic.

With some further but relatively minor revisions to this paper, I believe it would be a publishable article in IJERPH. I wish the authors all the best in their future work in this area.

Author Response

Thank you for the positive comments that “Overall, I believe the authors did a nice job of developing the manuscript throughout.” 

  1. Did the authors develop the semi-structured interview used in this study? Is this a validated measure? Please expand to further discuss this and establish its appropriateness for this study. 

Authors’ Response: We added information in section “2.2 Procedure” on page 4, lines 141-145 and lines 156-158.  

  1. How the authors arrived at their use of thematic analysis is not clear to me. Explain why this method was chosen over other available qualitative analysis approaches. 

Authors’ Response: We clarified this in section 2.3, lines 170-172. Thank you for the suggestion.  

Reviewer 3 Report

Dear authors, the article you are submitting is of interest for publication. However, the following clarifications are recommended:

- Introduction: It is advisable to start the introduction by talking about the importance of inclusive education and its practices/actions carried out in the formal setting. 

- References: It is advisable to incorporate some recent research (2021-2022) on inclusive education and sport.

- Research design: The use of interviews is described, but it should be noted that the research design is qualitative. It is also important that the authors comment on the rationale for this choice. 

- Data analysis: the analysis procedure is explained. However, there is no table with information on the dimensions and categories (definition of each of them), codes, frequency of occurrence, etc. It would be advisable to incorporate it. 

- Results: It is advisable to make a category map of the qualitative analysis carried out with NVivo software or an alternative program. A figure could be incorporated to help visualize the presentation of results later on. 

- Conclusions: It is advisable to expand a little more on the conclusions drawn from the study based on the objective. In addition, the conclusions section should include the limitations and prospects of the work presented. 

Author Response

  1. Introduction: It is advisable to start the introduction by talking about the importance of inclusive education and its practices/actions carried out in the formal setting.  

Authors’ Response: Thank you, the introduction now starts as you suggested it should, page 1, lines 28-32. 

  1.  References: It is advisable to incorporate some recent research (2021-2022) on inclusive education and sport 

Authors’ Response: We added recent studies on this topic on page 1, lines 31 (citation 1), 39 (citation 11), and 41 (citation 14).  

  1. Research design: The use of interviews is described, but it should be noted that the research design is qualitative. It is also important that the authors comment on the rationale for this choice.   

Authors’ Response: In the revised manuscript we note in the abstract that this is qualitative research and we discuss the rationale for qualitative research at the start of section 2.2 Procedure, on page 4, line 141-145.  

  1. Data analysis: the analysis procedure is explained. However, there is no table with information on the dimensions and categories (definition of each of them), codes, frequency of occurrence, etc. It would be advisable to incorporate it.  

Authors’ Response: We added a table with this information for the codes that informed our analysis, please see Table 3 on page 6 and explanation of coding process on page 5, lines 184-191.  

  1.  Results: It is advisable to make a category map of the qualitative analysis carried out with NVivo software or an alternative program. A figure could be incorporated to help visualize the presentation of results later on.  

Authors’ Response: We did not include a category map in the revision. After creating Table 3, we believe it captures a visual representation of results adequately so readers can understand the flow and rationale of the narrative presentation of results that follows. There were no comments by reviewers suggesting the results were difficult to follow, thus we think it is best to leave that section as is.  

  1. Conclusions: It is advisable to expand a little more on the conclusions drawn from the study based on the objective. In addition, the conclusions section should include the limitations and prospects of the work presented.  

Authors’ Response: Thank you for the suggestion, we did expand the conclusion (line 586-589) and moved the limitations and prospects of the work from the discussion to the conclusion (line 590-603).  

Round 2

Reviewer 1 Report

Thank you for resubmitting your manuscript entitled, When the normative is Formative: Partens' Perceptions of the Impacts of Inclusive Sports Programs. I was pleased to see that author(s) great effort to promptly address the reviewer's concerns.